# ARKitScenes: A Diverse Real-World Dataset For 3D Indoor Scene Understanding Using Mobile RGB-D Data

**Gilad Baruch** *  **Zhuoyuan Chen**   **Afshin Dehghan**   **Tal Dimry**   **Yuri Feigin**   **Peter Fu**

**Thomas Gebauer**   **Brandon Joffe**   **Daniel Kurz**   **Arik Schwartz**   **Elad Shulman**

**Apple**
arkitscenes@group.apple.com

## Abstract

Scene understanding is an active research area. Commercial depth sensors, such as Kinect, have enabled the release of several RGB-D datasets over the past few years which spawned novel methods in 3D scene understanding. More recently with the launch of the LiDAR sensor in Apple's iPads and iPhones, high quality RGB-D data is accessible to millions of people on a device they commonly use. This opens a whole new era in scene understanding for the Computer Vision community as well as app developers. The fundamental research in scene understanding together with the advances in machine learning can now impact people's everyday experiences. However, transforming these scene understanding methods to real-world experiences requires additional innovation and development. In this paper we introduce ARKitScenes. It is not only the first RGB-D dataset that is captured with a now widely available depth sensor, but to our best knowledge, it also is the largest indoor scene understanding data released. In addition to the raw and processed data from the mobile device, ARKitScenes includes high resolution depth maps captured using a stationary laser scanner, as well as manually labeled 3D oriented bounding boxes for a large taxonomy of furniture. We further analyze the usefulness of the data for two downstream tasks: 3D object detection and color-guided depth upsampling. We demonstrate that our dataset can help push the boundaries of existing state-of-the-art methods and it introduces new challenges that better represent real-world scenarios.

## 1 Introduction

Indoor 3D scene understanding is becoming key for many applications in the domains of augmented reality, robotics, photography, games, and real estate. More recently, modern machine learning techniques have fueled many state-of-the-art scene understanding algorithms. A variety of methods are addressing different parts of the challenge, like depth estimation, 3D reconstruction, instance segmentation [1, 2, 3], object detection [4, 5, 6, 7] and more. Most of these research works are enabled through a variety of real and synthetic RGB-D datasets that have been made available over the past few years [8, 9, 10, 11, 12, 13, 14, 15, 16]. Even though commercially available RGB-D sensors, like Microsoft Kinect, have made collection of such datasets possible, it is still

---

*Authors are listed in alphabetic order and contributed equally.

35th Conference on Neural Information Processing Systems (NeurIPS 2021) Track on Datasets and Benchmarks.

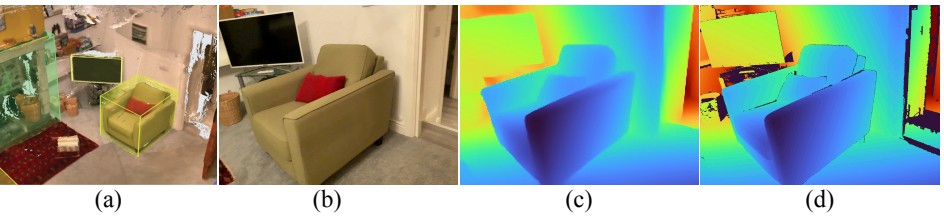

|     |     |     |     |
| --- | --- | --- | --- |
| (a) | (b) | (c) | (d) |

Figure 1: (a) Reconstructed mesh and annotated ground truth 3D bounding boxes. (b) A sample RGB frame. (c) A sample low-res (LR) depth frame. (d) A sample high-res (HR) depth frame.

not a trivial task to capture data at large scale with ground truth. In addition, almost all previous devices used for data collection is datasets such as SunRGBD [15] or ScanNet [14], are different from the hardware used by people nowadays. The lack of diversity in data and the gap in the depth sensing technology brings challenges in making the innovative research of the last decade practical for day-to-day use.

Recently, Apple released iPads and iPhones equipped with the LiDAR scanner [17]. It unleashed a new era in availability and accessibility of depth sensors. This work provides the first large-scale dataset that is captured with Apple's LiDAR scanner using handheld devices. It helps bridge the domain gap between existing datasets and widely available mobile depth sensors, and is the largest RGB-D dataset in terms of number of sequences and scene diversity collected in people's homes.

Our dataset, which we named *ARKitScenes*, consist of 5,048 RGB-D sequences which is more than three times the size of the current largest available indoor dataset [14]. These sequences include 1,661 unique scenes. Additionally we provide estimated ARKit camera poses as well as the LiDAR scanner-based ARKit scene reconstruction for all the sequences. A comparison of ARKitScenes with some of the existing datasets is shown in Table 1. In addition to the raw and processed data above, we provide high quality ground truth and demonstrate its usability in two downstream supervised learning tasks: 3D object detection and color-guided depth upsampling. For the 3D object detection task, ARKitScenes provides the largest RGB-D dataset labeled with oriented 3D bounding boxes for 17 room-defining furniture categories. ARKitScenes further uses high-resolution ground truth scene geometry that is captured with a professional stationary laser scanner (Faro Focus S70). We describe a technique used to register the high quality laser scans with mobile RGB-D frames captured with an iPad Pro. To our best knowledge, this is the first dataset that provides high quality ground truth depth data registered to frames from a widely available depth sensor. Finally, we evaluate the performance of state-of-the-art methods when trained and evaluated on ARKitScenes and highlight the challenges of existing methods in generalizing to real-world scenarios. In summary the contributions of this paper are as follows:

- We present ARKitScenes, the first RGB-D dataset captured with the widely available Apple LiDAR scanner. Along with the per-frame raw data (Wide camera RGB, Ultra Wide camera RGB, LiDAR scanner depth, IMU) we provide the estimated ARKit camera pose and ARKit scene reconstruction for each iPad Pro sequence.

- ARKitScenes is the largest indoor 3D dataset consisting of 5,048 captures of 1,661 unique scenes.

- We provide high quality ground truth of (a) depth registered with RGB-D frames and (b) oriented 3D bounding boxes of room-defining objects registered with the scene reconstructions.

- We demonstrate the effectiveness of the dataset in advancing state-of-the-art methods while highlighting the limitations of current methods and datasets in generalizing to realistic scenarios.

We expect ARKitScenes to stimulate the development of novel algorithms. Furthermore, we call for an evaluation and comparison of future work on ARKitScenes as it represents a diverse set of homes in the wild. And finally, we hope ARKitScenes bridges the gap between innovation and usability by the general public as it provides data captured with an RGB-D sensor which many people carry in their pockets.

| Dataset | Size | 3DOD Labels | HR #Frames | HR | LR |
|---|---|---|---|---|---|
| MPI-Sintel [25] | 35 scenes | - | 1,628 | 1024×436 | - |
| Middleburry [26] | - | - | 34 | 432×381 to 2964×2000 | - |
| NYU v2 [16] | 464 scans | 1,449 frames | - | - | - |
| SUN RGB-D [15] | 10k frames | 10k frames | - | - | - |
| SceneNN [23] | 100 scans | 100 scans | - | - | - |
| ScanNet [14] | 707 venues 1,513 scans | 1,513 scans | - | - | - |
| Matterport3D [22] | 2,056 rooms | 2,056 scans | 195k | 1280x1024 | - |
| **ARKitScenes** | **1,661 venues 5,047 scans** | **5,047 scans** | **450k** [2] | **1920×1440 Laser Scanner** | **256×192 ARKit Depth** |

Table 1: Overview of RGB-D datasets and their ground truth asset comparison with ARKitScenes. HR and LR represent High Resolution and Low Resolution, respectively.

## 2 Related work

Availability of large-scale datasets such as ImageNet [18] stimulated research, especially since supervised deep learning techniques re-gained popularity. In the context of scene understanding and 3D point clouds, several datasets were released in the past few years [16, 14]. These datasets have enabled a series of research investigations in various areas including semantic segmentation, object detection, room layout estimation, depth estimation and more. For outdoor scene understanding, several large scale datasets with a variety of scenes in real scenarios were released that have powered deep learning algorithms [19, 13, 20, 21]. However, when it comes to indoor scene understanding we are only limited to a few datasets [14, 15, 16, 22]. Outdoor and indoor datasets have very different characteristics, both caused by the size of the space and the type of sensors that are used to collect those datasets. Because of that, the methods designed for one will not necessarily perform the same in the other.

**Indoor scene understanding**. NYU v2 [16] is one of the earliest RGB-D datasets focusing on indoor scenes. It is composed of 464 scenes from three cities captured with a Kinect device. It also includes 1,449 densely labeled pairs of aligned RGB and depth maps annotated with 2D polygons. Sun RGB-D [15] further expands on previous work by introducing dataset with over 10,000 RGB-D frames along with 2D polygon and 3D bounding box labels. The labels are provided at frame level and not scene level, lacking view-point diversity. There are several other datasets which have focused on long indoor RGB-D captures [9, 14, 23, 24] to address the view-point diversity. Among these datasets ScanNet [14] is the largest and closest to ours in terms of scene diversity and assets. ScanNet provides 1,513 scans of 707 unique scenes along with dense 3D labels and CAD models. Our dataset on the other hand is three times the size of ScanNet, captured with Apple's LiDAR scanner instead of Kinect, and it provides 3D oriented bounding boxes and ground truth depth, which are not provided in [14].

**3D object detection** is a computer vision task that has gained a lot of popularity in recent years [27, 28, 29, 30, 31, 32, 33, 4, 5, 34, 7]. The majority of the published techniques focus on outdoor environments [6, 35] mainly in the context of autonomous driving, where diverse datasets such as [19, 13] exists. Most of these techniques make assumptions in the algorithm (e.g Bird's Eye View projection) that do not generalize well to indoor scenes. For indoor 3D object detection the number of datasets is limited. SunRGB-D and ScanNet are the two most commonly used ones. The former lacks scene level labels and the latter lacks oriented 3D bounding box labels. More

---

[2]High resolution ground truth depth maps are available for a subset of 2,257 scans over 841 different venues.

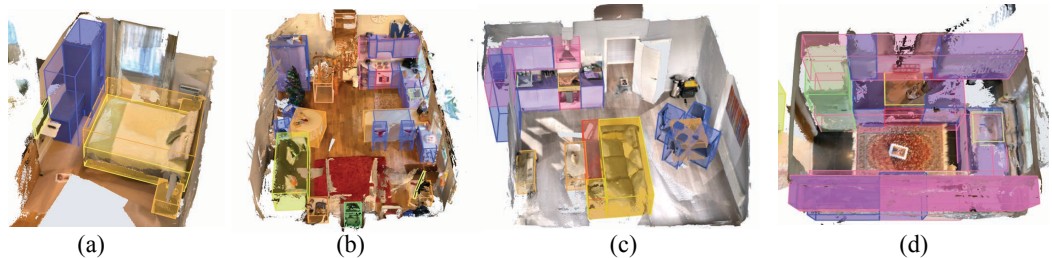

(a)       (b)       (c)       (d)

Figure 2: Examples of scene reconstructions and ground truth 3D bounding box annotations.

recent datasets such as [36] provide a variety of objects with 3D labels but do not provide depth sensor data. ARKitScenes provides the largest set of 3D oriented bounding boxes for a set of 17 room-defining object categories that addresses the gaps of previous indoor datasets.

**Color-guided depth upsampling** is the task of generating a high resolution (HR) depth map by using a high-resolution color image as guidance for the upsampling of a low resolution (LR) depth map [37, 38, 39]. HR depth maps are essential for many depth use cases which require high frequency depth information. Prior works are using datasets of a few images with high resolution ground truth, such as 34 images from Middlebury [40, 41, 26], or 58 synthetic images from MPI-Sintel[25], or using the low resolution depth sensors as HR ground truth [15, 16], and downscaling it even further in order to obtain the LR image. Table 1 compares those datasets and their properties. ARKitScenes is unique since it does not use the ground truth image as the source to generate the LR image by simple downscaling, instead it is providing LR depth maps captured with a consumer grade handheld LiDAR scanner and registered ground truth high-resolution depth maps captured with a professional stationary laser scanner. As a result upsampling methods trained with ARKitScenes are expected to generalize better to real-world scenarios as will be demonstrated below.

The rest of this paper is organized as follows. First, in Section 3, we introduce our data collection protocol as well as details around hardware and software used to capture the data. Moreover we cover the details around how we gather ARKitScenes ground truth. Next, in Section 4, we explore ARKitScenes for two downstream tasks of 3D object detection and depth upsampling. Finally, in Section 5, we summarize our findings and propose some future work.

## 3 ARKitScenes dataset

In this section, we describe the steps we pursued to acquire this dataset from collecting raw data in real-world homes, our data collection app, to fully-automatic spatial registration of the collected video sequences with highly accurate high-resolution stationary laser scans as well as manual annotation of 3D bounding box labels in the dataset.

### 3.1 Raw data acquisition

We used two main devices for data collection: The 2020 iPad Pro and Faro Focus S70. The 2020 iPad Pro is used to collect various sensor outputs such as IMU, RGB (for both Wide and Ultra Wide cameras) as well as the dense depth map from the LiDAR scanner via ARKit. We use the official ARKit SDK[3] to collect such information. Our data collection app runs ARKit world tracking and scene reconstruction during the capture. This is to provide live feedback to the operators, who are not computer vision experts, on tracking robustness and reconstruction quality. In addition to the handheld iPad Pro we utilized a Faro Focus S70 stationary laser scanner on a tripod to collect high-resolution XYZRGB point clouds of the environment.

For data collection locations, we use real-world homes which we rent for a full day. The home owners consented to this data being released publicly to facilitate research and development of indoor 3D scene understanding. The operator was instructed to remove any personally identifiable information prior to starting the captures. Data is collected in three major cities in Europe, London,

---

[3]https://developer.apple.com/documentation/arkit

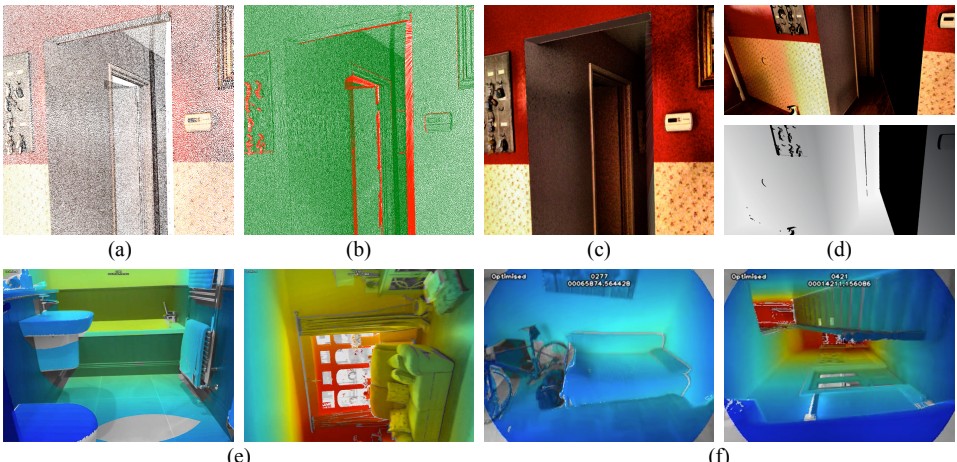

Figure 3: (a) Example XYZRGB point cloud from the laser scanner, (b) foreground triangle mesh (green) and occlusion triangle mesh (red), and (c) watertight textured mesh enabling rendering (d) color and depth buffers. (e) Exemplary iPad Wide and (f) Ultra Wide RGB frames with superimposed registered ground truth depth.

Newcastle, and Warsaw. To increase the indoor scene diversity and coverage we took two criteria into account when selecting homes for data collection: the socioeconomic status (SES) of the household as well as the location of the house in the city. The houses in our dataset are selected from rural, suburban, and urban location in each of the aforementioned cities. Additionally we included houses from all three tiers of low, medium, and high SES levels.

After selecting the house for data collection, we divide each house into multiple scenes (in most cases, each scene covers one room), and perform the following steps. First, we use a Faro Focus S70 stationary laser scanner on a tripod to collect highly accurate XYZRGB point clouds of the environment. Tripod locations are chosen to maximize surface coverage, and on average we collected four laser scans per room to ensure good coverage. Second, we record up to three video sequences attempting to capture all surfaces in each room using the iPad Pro. Each sequence follows a different motion pattern and captures the ceiling, floors, walls, and room-defining objects. The on-device ARKit world tracking poses as well as the scene reconstruction are stored and provided with the dataset, and they are also overlaid on the camera stream in the data collection app, to ensure the objects in the room are well covered.[4]

Throughout the collection of all data, we attempt to keep the environment completely static, i.e. we make sure no objects move or change their appearance. However, since data collection of a venue takes an average of six hours and many venues are lit by sunlight, the lighting situation can change during that time resulting in potentially inconsistent illumination between the different sequences and scans.

### 3.2 Ground truth generation

**Ground truth poses and depth maps.** After data collection, in a one-time offline step, we first spatially register all XYZRGB point clouds from the stationary laser scanner into a common coordinate system using the proprietary software Faro Scene, which for most scenes fully automatically estimates a 6DoF rigid body transformation for each scan transforming it into a common *venue coordinate system.* Note that a venue (usually a house or apartment) can comprise multiple unique scenes. After this step, and throughout the rest of this paper, the XYZRGB point clouds are always assumed to be expressed in common venue coordinates.

Our approach to estimate the ground truth 6DoF pose of the iPad Pro's RGB cameras with respect to the venue coordinate system requires the generation of synthetic views of our laser scan of the venue. Rendering these XYZRGB point clouds from novel viewpoints poses unique challenges. In particular, we require that far geometry is correctly occluded by near geometry and that geometry is discarded for which a direct line-of-sight from the novel view point cannot be guaranteed (e.g.

---

[4]An example of such scan patterns and our app UI is shown in our supplementary materials.

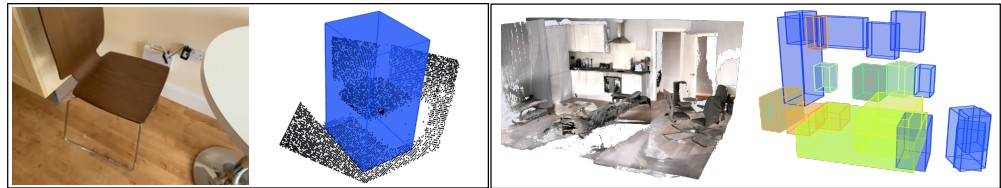

Figure 4: An example of VoteNet [4] prediction output for single-frame and whole-scene detection.

they might be occluded by surfaces that were not captured in the scan). Naively rasterizing the scans as unstructured point clouds would violate both of these requirements.

Instead, we first find for each scanned 3D point cloud a triangulation by reducing it to two dimensions via stereographic projection with respect to the laser scanner's nodal point and computing a 2D Delaunay triangulation. When applied to the 3D point cloud this triangulation is forming a watertight mesh, for which we compute texture coordinates referring to an equirectangular texture into which we write the RGB color information from the XYZRGB point cloud.

The triangles of this mesh are then split into two sets by applying a threshold on the angle between the triangle normal and the ray from the triangle center to the laser scanner nodal point. When this angle exceeds a threshold the triangle is considered to manifest a discontinuity and will be used as *occlusion geometry*; otherwise it will be used as *foreground geometry*. This separation enables reasoning about unobstructed line-of-sight. See Figure 3 (b) for an example visualization.

Finally, the two sets of triangles are rendered in separate passes using OpenGL, both writing to the depth buffer, while only front-facing triangles of the foreground geometry write a nonzero value to the stencil buffer. In all other cases the stencil buffer is cleared. As a result only fragments with unobstructed line-of-sight towards front-facing foreground geometry will have a nonzero stencil value, hence the stencil buffer can be used to mask pixels in the rasterized output. To create a joint rendering of multiple scans each scan is rendered separately and the renderings are merged in screen space using the individual depth and stencil buffers. After repeating this process for every scan the rasterization of a synthetic view is complete. This strategy is independent of the order in which individual rasterization results are merged and enables rasterizing depth maps and RGB images of one or multiple stationary laser scans from arbitrary viewpoints.

To be used as ground truth, the next objective is to determine the 6DoF pose of the handheld iPad Pro's cameras with respect to the venue coordinate system. To this end we extract local image features and descriptors in *keyframes* of the camera sequence and store them as query features. Using the presented rasterization method we then create renderings of the laser scans, in which we detect and describe the same kind of local image features and store them as reference features together with their 3D locations, which can be looked up in the rasterized depth maps. We then match each query feature with the most similar reference feature leading to a set of 2D-3D correspondences for each keyframe. We use RANSAC [42] and PnP to estimate initial camera poses using these sparse correspondences. To improve over these initial estimates we jointly solve for the camera poses of all keyframes that minimize a dense photometric error metric. For a single keyframe the metric optimizes photometric consistency between (1) the keyframe image and rendered views of the laser scans and (2) the keyframe image and projected views of neighboring keyframes that share visibility of the same parts of the laser scan surface geometry. This registration is performed once offline per video sequence.

When the refined ground truth poses are determined we render camera frame-aligned ground truth depth maps encoding per-pixel orthographic depth. This dense ground truth depth map, along with the per-keyframe ground truth pose are provided as part of the dataset. Figure 3 visualizes the results for a set of keyframes from the dataset.

**3D object bounding boxes**. We use a custom tool to manually annotate 3D oriented bounding boxes for 17 categories of room-defining furniture. The annotation happens on the ARKit scene reconstruction, which leads to a colored mesh of the scene. Additionally, our labeling tool allows annotators to see real-time projections of 3D bounding boxes onto video frames to facilitate accurate annotation.

**Train/Validation/Test split.** We split the venues of ARKitScenes into 80% for training, 10% for validation and the remaining 10% are a held-out test set that we do not release. The training and

| Categories | Cabinet | Refrigerator | Shelf | Stove | Bed | Sink | Washer | Toilet | Bathtub |
|---|---|---|---|---|---|---|---|---|---|
| VoteNet [4] | 0.371 | 0.627 | 0.124 | 0.003 | 0.850 | 0.311 | 0.453 | 0.755 | 0.933 |
| H3DNet[43] | 0.402 | 0.594 | 0.100 | 0.016 | 0.882 | 0.401 | 0.490 | 0.838 | 0.930 |
| MLCVNet [30] | 0.451 | 0.700 | 0.169 | 0.024 | 0.880 | 0.402 | 0.515 | 0.859 | 0.941 |

| Categories | Oven | Dishwasher | Fireplace | Stool | Chair | Table | TV/Monitor | Sofa | Overall |
|---|---|---|---|---|---|---|---|---|---|
| VoteNet [4] | 0.183 | 0.029 | 0.221 | 0.030 | 0.201 | 0.310 | 0.006 | 0.683 | 0.358 |
| H3DNet[43] | 0.241 | 0.039 | 0.195 | 0.088 | 0.252 | 0.322 | 0.015 | 0.704 | 0.383 |
| MLCVNet [30] | 0.242 | 0.030 | 0.385 | 0.080 | 0.315 | 0.366 | 0.041 | 0.719 | 0.419 |

Table 2: Evaluation of whole-scene detection tasks on our ARKitScenes dataset.

| Categories | Cabinet | Refrigerator | Shelf | Stove | Bed | Sink | Washer | Toilet | Bathtub |
|---|---|---|---|---|---|---|---|---|---|
| VoteNet [4] | 0.421 | 0.468 | 0.168 | 0.047 | 0.777 | 0.567 | 0.418 | 0.761 | 0.779 |
| H3DNet[43] | 0.468 | 0.368 | 0.170 | 0.060 | 0.713 | 0.584 | 0.353 | 0.721 | 0.774 |
| MLCVNet [30] | 0.113 | 0.188 | 0.085 | 0.003 | 0.652 | 0.365 | 0.205 | 0.603 | 0.657 |

| Categories | Oven | Dishwasher | Fireplace | Stool | Chair | Table | TV/Monitor | Sofa | Overall |
|---|---|---|---|---|---|---|---|---|---|
| VoteNet [4] | 0.324 | 0.107 | 0.476 | 0.167 | 0.482 | 0.436 | 0.150 | 0.705 | 0.427 |
| H3DNet[43] | 0.302 | 0.069 | 0.323 | 0.174 | 0.505 | 0.431 | 0.172 | 0.662 | 0.403 |
| MLCVNet [30] | 0.173 | 0.025 | 0.140 | 0.080 | 0.366 | 0.329 | 0.024 | 0.591 | 0.271 |

Table 3: Evaluation of per-frame detection tasks on our ARKitScenes dataset.

validations set include the $5,048$ sequences which we release. Since the split is determined on a per-venue basis, all laser scans and iPad sequences of a given venue fall into the same bin. The split is common across all downstream tasks including those discussed below.

## 4  Tasks and benchmarks

To evaluate the performance of different algorithms on ARKitScenes, we chose two computer vision tasks and trained state-of-the-art machine learning models using our dataset.

### 4.1  3D object detection

**Problem details.** As a fundamental task in computer vision, the goal of 3D object detection is to localize and recognize objects in a 3D scene. In ARKitScenes, we focus on two setups for our 3D object detection: *single-frame* based and *whole-scene* based. Given an RGB-D video sequence, the former targets detecting objects in each single RGB-D frame, while the latter detects objects from the whole reconstruction of the 3D scene.

**Ground Truth Processing.** Given that our ground truth bounding boxes are labeled on each scene reconstruction, we can directly use them for *whole-scene* 3D object detection scenario. However, for *single-frame*, we need to pre-process them as follows. Similar to outdoor benchmarks such as KITTI [19], we only keep boxes that at leasat have five corners in the camera frustum. Additionally a bounding box is removed if it contains fewer than 10 points.

After the bounding box exclusion, 65% of frames will be empty with no bounding boxes remaining. Moreover, we keep at most 300 frames for each video scan to down-weight really long videos. After filtering and long video sampling, we are left with over one million frames. Training current state-of-the-art detection algorithms off the shelf with this many frames will take a very long time. To further speedup, we subsampled one third of the data for our training, ending up with 365,007 frames: 323,868 for training and 41,139 for testing.

**Models and training details.** Building on the PointNet++ backbone and Hough voting modules, VoteNet [4] achieves state-of-the-art performance on indoor scenarios. Along this line, MLCVNet [30] and H3DNet [43] further improves the VoteNet model by leveraging an attention model and extra geometric primitive prediction.

We followed the original design of these three approaches: the backbone network is a PointNet++ with several set-abstraction layers and feature propagation (upsampling) layers with skip connections, which outputs a subset of the input points with XYZ and an enriched $C$-dimensional feature vector. The results are $M$ seed points of dimension $(3 + C)$. Each seed point generates one vote. Each seed goes through a Hough voting module with supervision to guide each foreground

| Factor | $\ell_1$ | | | | | RMSE | | | | |
|---|---|---|---|---|---|---|---|---|---|---|
| | Bilinear | JBU | FGI | MSG | MSPF | Bilinear | JBU | FGI | MSG | MSPF |
| x2 | 0.0251 | 0.0256 | 0.0253 | 0.0176 | **0.0153** | 0.0424 | 0.0426 | 0.0435 | 0.0376 | **0.0369** |
| x4 | 0.0250 | 0.0255 | 0.0255 | 0.0175 | **0.0149** | 0.0423 | 0.0424 | 0.0434 | 0.0373 | **0.0362** |
| x8 | 0.0254 | 0.0259 | 0.0260 | 0.0185 | **0.0152** | 0.0443 | 0.0436 | 0.0443 | 0.0383 | **0.0363** |

Table 4: $\ell_1$ and RMSE for Depth Upsampling methods over ARKitScenes

point to vote to its bounding box center. Finally a last proposal module aggregates the votes with a shared PointNet to predict center, size and category of each bounding box. We use ADAM optimizer to train our model for 200 epochs, with learning rate 0.001 and decay rate 0.1 at the 80-th and 120-th epoch. We augment our data with rotation, scaling and translation.

For single frame evaluation, training the vanilla version of all these models will take a very long time to converge. To speedup, we train all three models with constrained computational budget (at most two weeks) by using a lighter network backbone and fewer training epochs. First, as all three models are based on PointNet++[44] backbone with four set abstraction (SA) layers and two feature propagation/upsamlng (FP) layers, we reduced the output dimension of four SA from 256 to 128 and the depth of each MLP module from three layers to two layers. Second, we reduced the training epoch number from 180, 360, 360 to 100, 80, 60 for VoteNet [4], MLCVNet [30] and H3DNet [43] respectively. Each of the three models training take about approximately two weeks after our speedup.

**Benchmark.** As a baseline evaluation, we first show the performance of object detection on whole-scene in Table2. VoteNet[4] is able to achieve mAP (mean average precision) of 0.358, while extra primitive supervision [43] and attention model [30] can further improve the overall performance to 0.383 and 0.419 respectively. We observe that these models perform better on large objects, such as refrigerator and bathtub, and struggles on small objects such as stove, dishwasher and TV/monitor. In Table 3, we show performance of single-frame detection. We observe a similar trend for different categories in both task settings. Finally, Fig 4 shows qualitative results of VoteNet on ARKitScenes for both *single-frame* and *whole-scene*. For more results and additional experiments please refer to the supplementary material.

## 4.2 Color-guided depth upsampling

**Problem details.** Depth upsampling is a common approach used to enhance low resolution (LR) depth maps to a high resolution (HR), higher fidelity depth map using an HR color image as guidance. HR accurate depth maps are imperative for downstream tasks such as 3D reconstruction, augmented reality, and photography. All of these require high frequency depth information which is often lost at low resolution.

Initial classical approaches to color-guided depth upsampling include both optimization and filtering-based methods [38, 39]. These approaches performed relatively well but are often hand-crafted and lack the ability to capture global structure and context. More recently, a data-driven approach using deep neural networks helps overcome some of these challenges. One of the prominent works in this area is *Multi-Scale Guided Networks* (MSG) [37], an encoder-decoder network extracting features at different resolutions from the guiding image in the encoder branch, and concatenating it in the corresponding resolution of the decoder branch of the depth map. The network is trained to learn the differential correction of the naïve upsampling. The current state-of-the-art in the field of guided depth upsampling is *Multi-Scale Progressive Fusion* (MSPF) [45], where the authors suggest the use of two different encoder branches, one for depth and one for color, along with a reconstruction branch that applies fusion blocks to restore the HR depth map.

**ARKitScenes adaptations.** As mentioned in Section 2, prior works on depth upsampling were demonstrated over LR depth maps that were generated by down-sampling *ground truth* HR depth maps, sometimes with the addition of artificial noise. This inherently makes these datasets easier for processing but limits the evaluation to non-realistic scenarios in which the low resolution sensor is only suffering from synthetic artifacts not necessarily representative of real-world scenarios. However, ARKitScenes brings a more realistic challenge of upsampling a low resolution depth map captured with the LiDAR scanner on a mobile device that has artifacts inherent to active sensing. Hence the challenge becomes twofold: **depth upsampling and depth artifacts correction**.

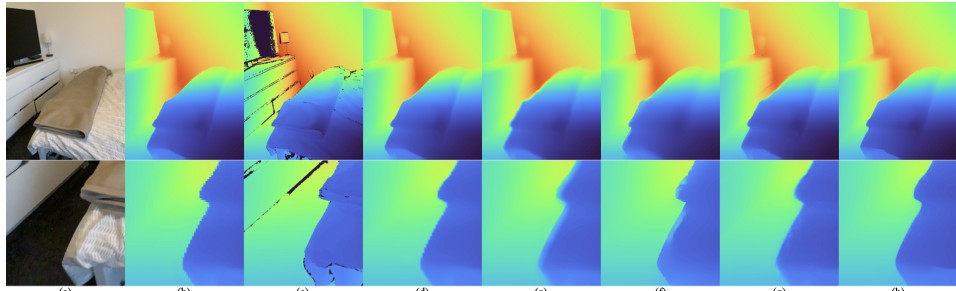

Figure 5: Examples of upsampling an image (top row) by a factor of 8 and a zoom-in on the bottom left of the image (bottom row), showing: (a) HR color, (b) LR ARKit depth map, (c) HR ground truth and the results of the different upsampling methods: (d) Bilinear, (e) JBU, (f) FGI, (g) MSG and (h) MSPF.

Another topic is that the HR ground truth depth map in ARKitScenes is a projection of laser scans that were taken from different viewpoints, therefore occlusions may cause some parts of the image to lack depth information. Hence, some of the methods in the existing research need to be adapted to handle the special value of no-depth pixels in the ground truth. Specifically, the *Structural Similarity Index Measure* (SSIM) loss used by MSPF [45] cannot be used over ARKitScenes, as it is a *full-reference method*, requiring the information in all pixels without masking. In addition, the edge loss used by MSPF operates on the entire image and therefore needed to be changed to a more robust loss. We opted to use the edge loss suggested by [46]. More details about these adaptations and experiments can be found in the supplementary material.

**Experimental results.** We would like to compare the results of existing methods on ARKitScenes. For these experiments we reproduced three classical approaches for depth upsampling - naïve *Bilinear* interpolation, *Joint Bilateral Upsampling* (JBU) [38] and *Fast Guided Global Interpolation* (FGI) [39], as well as the two mentioned modern DNN-based solutions - MSG [37] and MSPF [45].

In order to have cleaner data for training, we removed frames where regions of missing depth information took more than 40% of the HR depth map. Also, in order to avoid issues originating from the ground truth registration process, we ignore frames at which the *Root Mean Square Error* (RMSE) between the LR and a downscaled HR depth map is more than 7cm or when comparing per pixel, more than 20% of the pixels in the frame differ by more than 5cm.

With a target to improve run-time while maintaining high diversity between frames, we sub-sampled the dataset by taking a single frame every two seconds. This led us to train the models with 39k frames from the train split and evaluate on a different 5.6k frames from the validation split. The validation split was further manually filtered to include only frames without issues arising from depth aggressors that are hard to detect automatically such as specular or transparent objects. The train and validation split were taken from completely different houses. More details on the experiments settings along with more examples can be found in the supplementary material. A visual comparison shown in Figure 5, along with many more examples in the supplementary material clearly show how the methods that were trained on ARKitScenes (MSG and MSPF) produce sharper edges and more realistic structure in the predicted image. It is also supported by comparing the absolute difference ($\ell_1$) and the *Root Mean Square Error* (RMSE) between the methods on Table 4, in which MSG and MSPF outperform the classical methods with a high margin. While those results are encouraging, we believe that future research could leverage this new dataset to suggest more sophisticated methods benefiting from the quantities and nature of this dataset to overcome the difficulties in upsampling a noisy LR image in real-world scenarios.

## 5    Conclusions

We presented ARKitScenes, it is not only the first dataset that is captured with Apple's LiDAR scanner, but also to the best of our knowledge the largest indoor RGB-D dataset ever collected with a mobile device. We showed how our dataset can be used for two downstream computer vision tasks of 3D object detection and color-guided depth upsampling. We believe ARKitScenes will enable the research community to push the boundaries of existing state of the art and develop technologies that better generalizes to real-world scenarios.

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
