# OpenReview forum: "ARKitScenes: A Diverse Real-World Dataset For 3D Indoor Scene Understanding Using Mobile RGB-D Data"
_NeurIPS.cc/2021/Track/Datasets_and_Benchmarks/Round1 — NeurIPS 2021 Datasets and Benchmarks Track (Round 1)_

### Official Review · Reviewer_Pheb · 2021-07-04
**Great dataset lacking enough experiments**

**Rating:** 7
**Confidence:** 5
**Clarity:** The paper is well written.

**Strengths:**

- This is a good dataset I'm very glad to see. Right now a lot of 3D understanding tasks are bottlenecked be 3D datasets since 3D annotations are very hard to collect. Reconstructing 5000 rooms is very challenging and I would expect this to encourage multiple follow-ups.
- The supp video is great and demonstrates the high quality of the dataset. I'm very concerned about the reconstruction quality before looking at the video. Qualitatively it looks a very high-quality 3D dataset, and way better than Matterport3D and ScanNet. I'm very excited about that!
- The overall writing is good.

**Weaknesses:**

- My main concern is the paper does not have enough experiments to demonstrate the quality or usefulness of the dataset. In the abstract I see "We demonstrate that our dataset can help push the boundaries of existing state-of-the-art methods". However, the paper neither includes any discuss or evaluation of the reconstruction quality, nor provide any experiments to show generalization. All downstream tasks, including 3D object detection and depth upsampling, are conducted on the proposed dataset. To validate the quality of the dataset further, I would like to see how the model trained on the proposed dataset can generalize to other RGB-D  datasets (Matterport3D, ScanNet, Replica, etc.). I totally understand it may not generalize well and the problem may be the quality of other datasets, but it is important to show, so that we can understand how well we're doing.
- ARKit looks a little bit oversold in the paper. When I saw ARKit I would expect the dataset is constructed directly from a iPhone/iPad and I can do the same reconstruction using my apple device. As said in the abstract, "This opens a whole new era in scene understanding for the Computer Vision community as well as the developers." However, the spatial reconstruction first comes from a Faro Focus S70 0 stationary laser scanner sensor on four sparse locations of the room, which looks the same as Matterport3D. And I think the ARKit is only used to refine reconstruction details of the dataset. Why is a lidar sensor necessary? If we do not need it, we can scale the approach to even more scenes.



**Additional Feedback:**

There are minor points.

- The paper cites Matterport3D but does not include it in the comparison (Table 1). To my understanding, Matterport3D has 2056 rooms, which is larger than ScanNet. So it is worth including it.


**Correctness:**

The dataset is constructed in a sound way. Overall, there are three parts:

1. Raw Data Acquisition. The paper uses a LiDar sensor and ipad pro depth sensor.
2. Ground truth generation. The paper spatially register the point cloud from the lidar sensor which is standard procedure. They also propose a new method to efficiently rasterize multiple point clouds. Then they register the ipad pro camera pose using SIFT-like feature matching. It sounds solid and doable to me.
3. Manual annotation of 3D Object Bounding Boxes. This follows the standard procedure in the area.

**Documentation:**

- There are sufficient details on data collection in the man paper section 3.
- There is a url for a subset (2 scans) of the dataset. It makes sense to me given that this is an extremely large datasets. With 5000 scans it can easily be hundreds of GBs or several TBs. I've took a look at the data and it looks as promised in the paper.
- I do not find a hosting, licensing and maintenance plan. If I miss it, please let me know.



**Ethics:**

According to the paper,

- they rent real-world homes for a full day.
- the homeowners has consented to public research usage of the data.
- the operators have been instructed to remove all personally identifiable information.

I do not have ethical concerns about the dataset.

**Relation To Prior Work:**

Table 1 summarizes the comparison with other RGB-D datasets. As discussed in the paper,

- This is the first dataset using Apple ARKit, which actually uses a IPad Pro. Please see my concerns in Weaknesses.
- This is the largest indoor RGB-D dataset. To the best of my knowledge this is true.

**Summary And Contributions:**

The paper proposes the first dataset that is captured with Apple LiDAR sensor and also the largest indoor RGB-D dataset, which include over 5000 rooms. Experiments of 3D Object Detection and Color Guided Depth Upsampling are shown as downstream tasks.

---

### Official Review · Reviewer_1QjY · 2021-07-05
**Large scale indoor RGB-D dataset using Apple's new sensor, but only two existing tasks.**

**Rating:** 4
**Confidence:** 4
**Correctness:** It is ok.
**Clarity:** It is well written.

**Strengths:**

First of all, the writing is well-organized, and the reader can easily follow the logic of the paper. The author presents clear data collection and processing in the paper and the comparison table1 obviously states the strength of their dataset. The proposed dataset is much larger than the size of the previous SOTA RGB-D dataset, ScanNet. This dataset also contains useful information such as high-quality ground truth depth, and oriented labeled bounding box. The data collecting method used in the paper is mobile device-based method which could be easily applied to different fields.

**Weaknesses:**

1. For both of the two tasks, the baselines are not very efficient enough to pass the bar for a neurips dataset paper. Moreover, to my surprise, the authors only proposed two tasks on this dataset, when the raw data itself could be more useful than just object detection and upsampling.

2. The author claims the scene diversity is one of the most innovative parts compared with precious work. However, the authors don’t provide a quantitive comparison or statistical evidence to support their claim.

3. The author claim that their dataset highlights the challenges of existing methods in generalizing to real-world scenarios. However, I cannot find any evidence to support the stated challenges where the generalization ability of the existing methods is poor on this dataset.

**Additional Feedback:**

While the proposed ARKitScenes dataset provides much larger indoor RGB-D data with high-quality depth and labeled oriented bounding box information than previous work. Some of the claims in the paper don’t have sufficient experiments and quantitative supports. The benchmarks are not enough for a dataset paper. Meanwhile, the authors don’t provide some of the required
information in the supplementary.

**Documentation:**

The author did not provide the dataset document, URL, or webpage to the dataset, author statement, and license for the dataset which is required in the supplementary material.


**Ethics:**

No concerns for this dataset.

**Relation To Prior Work:**

It is ok.

**Summary And Contributions:**

This paper proposes a large indoor scene RGB-D dataset which is collected using the new Apple LiDAR scanner. This dataset consists of whole-scene-level and single-frame-level data with high-quality ground truth depth, and oriented labeled bounding box for furniture objects in each scene. This paper demonstrates the usefulness of the data by performing two tasks: 1) 3D object
detection and RGB-D guided upsampling. The authors encourage further research on more complicated methods for these two tasks with the helping of the high quality of this dataset.

---

### Official Review · Reviewer_7GW1 · 2021-07-06
**Initial Review for ARKitScenes**

**Rating:** 6
**Confidence:** 2
**Correctness:** Yes.
**Clarity:** Yes.

**Strengths:**

I believe the dataset will be a very promising resource for researchers in 3D indoor scene understanding. By using the latest commercial sensors, high-resolution data is available. In addition, it seems like the data is collected from real-world scenarios in daily life, which makes more connections from research to daily applications.

**Weaknesses:**

1. The biggest drawback of the proposed dataset is the motivation. Given 3D scene understanding is a well-studied area in computer vision community, the proposed dataset is not novel in terms of problem definition and formulation. After reading the entire paper, it seems like the most important key factor is the usage of the latest apple LiDAR and the availability of the data resources. Although the contribution of providing high-resolution data is valid, the work itself is somehow incremental.

2. There should be lots of SOTAs in 3D scene understanding. Only few of them are provided. Is it possible to provide more experiments?

**Additional Feedback:**

See weakness and ethics

**Documentation:**

Yes.

**Ethics:**

It may not be a valid academic concern but it is valuable to discuss. Given the data is collected from mobile devices, is it possible that the data collection process may violate personal privacy?

**Relation To Prior Work:**

Yes.

**Summary And Contributions:**

This paper presents a large-scale dataset for 3D indoor scene understanding. The data is collected by a dedicatedly designed sensor (apple LiDAR). High-resolution data is provided. A Benchmark of some SOTA models are provided.

---

### Official Review · Reviewer_o7aa · 2021-07-06
**3D Indoor Scene Understanding dataset using laser scans, lidar and rgb - useful for the community but some details missing/unclear**

**Rating:** 6
**Confidence:** 2
**Clarity:** The paper is well written and easy to…

**Strengths:**

- The dataset is valuable for the computer vision community.
- The dataset will be publicly available (note: only training and validation but not test, the reason is not specified).
- The dataset may cover a gap in the literature, especially for the color-guided depth upsampling task as, according to the authors, other datasets use the high-quality ground truth image to generate a low-resolution image by downscaling; in contrast, the low-resolution content provided with ArKitScenes is captured with a consumer grade sensor, thus incorporating noise present in the sensor.  The dataset may also be useful for the 3D object detection task as a complement to other existing datasets.
- Supplementary material is really helpful to better understand some parts of the paper (even though the main paper should be self contained).


**Weaknesses:**

- It is unclear why the 2 tasks included have been selected. Do the authors envision the dataset being used for other computer vision tasks?
- It is unclear why the methods used for benchmarking have been selected.
- Test set will not be made publicly available. The authors do not specify why - will there be a public challenge using this data?
- Some data collection and data curation details or design choices are either left out or not clearly explained/justify (explained in the Correctness and Documentation questions). This is the main reason for my rating.


**Additional Feedback:**

Suggestions for improvement:
- Some tables and figures, for instance Table 2, appear before their reference in the text, their position should be changed.
- Table 1: It would be useful to add the paper references in the Dataset column.
- Line 25: add references for depth estimation and 3D reconstruction (same as for instance segmentation and object detection).
- The paper repeats a handful of times that some details are given in the supplementary material. This could just be said once to have more space for more important content.

Questions:
- In Section 3.2, what's the intuition behind dividing into occlusion geometry and foreground geometry?

**Correctness:**

The evaluation appears to be performed correctly. Some discussion of the results would be helpful to better understand the possible limitations of the dataset. For instance, L279-L291, what could be the reason for this behaviour? Could that be due to a limitation/defect of the dataset, or is it just method-dependent?

I have some doubts regarding how the dataset has been constructed:
- The scanners used may be affected by sunlight. According to L157-159, the lighting situation changed throughout the scans of a single scene due to the time needed to capture the scene. If the scanners are indeed affected by sunlight, the quality of the scans may vary a lot. Was this accounted for somehow?
- To select households, their socioeconomic status was considered. How was this information obtained?
- The stationary scanner location is of utmost importance and impacts the subsequent data curation process (i.e. Section 3.2). However, the only details re. how these locations are selected are given in L149-150 (chosen to maximize surface coverage, collecting 4 scans per room on average). What do authors mean exactly by maximizing surface coverage? Why not doing more scans to really maximize the coverage and minimize occlusions?
- L188-197, it looks like the first selected scan impacts the quality of the final merged render - as I understand from the paper, the triangles classified as occlusion geometry from the render of the first scan will not be modified by any of the subsequent scans used to render a specific view, even if those scans contain information that is correct on the locations of the first scan's occlusion geometry. What is the intuition behind that? Is there any protocol to select the first scan?
- Overall Section 3.2 is a bit unclear or difficult to follow. For instance, in L203 it is said that using the afore-described rasterization method the LiDAR scenes are rendered, but as I understand from the previous text, that process was to render arbitrary viewpoints from the stationary laser scans.
- Re. 3D Object bounding boxes creation. Projections of 3D bounding boxes are indeed useful to visually check the quality of the annotation, however, they are not error-free - i.e a few degrees of error will not be easily caught at first sight. Since most of the objects are parallel/perpendicular to other surfaces which could be automatically recovered, they could be used to give a prior for the 3D bounding box orientation. For instance, the floor plane would be really helpful to fix the 3D orientation of objects that lie on the floor or parallel to the floor. Same for walls. Did you try other inspection approaches like this one?

**Documentation:**

- Sufficient detail on data collection (see some questions above).
- Technical documentation of the dataset contents is missing.
- An URL to sample data and scripts to run the evaluations is provided.
- No details regarding hosting nor maintenance have been found. Details re. license are found in the checklist only.
- Intended uses are included in the paper.

**Ethics:**

No ethical concerns are envisioned.

**Relation To Prior Work:**

Yes.

**Summary And Contributions:**

The paper presents ARKitScenes, a dataset of 3D indoor scenes, captured using a stationary laser scan and RGB+D (with Apple's LiDAR scanner). The dataset provides the raw data, camera pose, surface reconstruction for each scene, registered RGB-D frames and oriented bounding boxes of particular objects. According to the authors, it is the largest in terms of image captures and scenes.  The paper explains the data collection and registration process and provides a benchmark for 2 specific tasks, 3D object detection in single frames and scenes, and color-guided depth upsampling.

---

### Decision · Program_Chairs · 2021-07-27

**Decision:**

Accept

**Comment:**

The provided dataset of indoor 3D data shows to be the largest up to date of its kind, including a large variability in content. There are different opinions regarding this contribution, in particular about the novelty of the tasks and its evaluation. However, the authors clearly addressed during rebuttal period the main dataset contributions. I overall agree that the dataset is going to be very a very useful resource for the community.